# Comparison between CMIP5 and CMIP6 Models over MENA Region Using Historical Simulations and Future Projections

**Mohammed Magdy Hamed** [1,2,*], **Mohamed Salem Nashwan** [3], **Mohammed Sanusi Shiru** [4] **and Shamsuddin Shahid** [2]

1. Construction and Building Engineering Department, College of Engineering and Technology, Arab Academy for Science, Technology and Maritime Transport (AASTMT), B 2401 Smart Village, Giza 12577, Egypt
2. Department of Water and Environmental Engineering, School of Civil Engineering, Faculty of Engineering, Universiti Teknologi Malaysia (UTM), Skudia 81310, Johor, Malaysia
3. Construction and Building Engineering Department, College of Engineering and Technology, Arab Academy for Science, Technology and Maritime Transport (AASTMT), Elhorria, Cairo P.O. Box 2033, Egypt
4. Department of Environmental Sciences, Faculty of Science, Federal University Dutse, P.M.B, Dutse 7156, Nigeria
* Correspondence: eng.mohammedhamed@aast.edu

**Abstract:** The study evaluated the ability of 11 global climate models of the latest two versions of the Coupled Model Intercomparison Project (CMIP5 and CMIP6) to simulate observed (1965–2005) rainfall, maximum ($T_{max}$) and minimum ($T_{min}$) temperatures, mean eastward (uas) and northward (vas) wind speed, and mean surface pressure. It also evaluated relative uncertainty in projections of climate variables using those two CMIPs. The European reanalysis (ERA5) data were used as the reference to evaluate the performance of the GCMs and their mean and median multimodel ensembles (MME). The study revealed less bias in CMIP6 GCMs than CMIP5 GCMs in simulating most climate variables. The biases in rainfall, $T_{max}$, $T_{min}$, uas, vas, and surface pressure were −55 mm, 0.28 °C, −0.11 °C, −0.25 m/s, −0.06 m/s, and −0.038 Kpa for CMIP6 compared to −65 mm, 0.07 °C, −0.87 °C, −0.41 m/s, −0.05 m/s, and 0.063 Kpa for CMIP5. The uncertainty in CMIP6 projections of rainfall, $T_{max}$, $T_{min}$, uas, vas, and wind speed was relative more narrow than those for CMIP5. The projections showed a higher increase in $T_{min}$ than $T_{max}$ by 0.64 °C, especially in the central region. Besides, rainfall in most parts of MENA would increase; however, it might decrease by 50 mm in the coastal regions. The study revealed the better ability of CMIP6 GCMs for a wide range of climatic studies.

**Keywords:** GCM; climate change; uncertainty; coupled model intercomparison project; seasonal variability





## 1. Introduction

Climate change is a concern due to its potentially catastrophic consequences [1]. It has caused worldwide temporal and spatial changes in climate variables [2–6]. It has also affected the frequency and severity of natural disasters, including droughts [7–11], heatwaves [12,13], and flooding [14–16]. Numerous industries have also been impacted by climate change, including water resources [17–19], agriculture [20–22], energy [23,24], and health [25,26]. Project changes in climate disaster-prone areas are vital for planning climate change adaptation and mitigation [1,27]. Global climate models (GCMs) can mimic the impact of greenhouse gas (GHG) emissions on climate systems and correctly anticipate future circumstances based on this information [28,29]. Public access to these GCMs has been granted through the Coupled Model Intercomparison Project (CMIP) via https://esgf-node.llnl.gov/ accessed on 1 July 2022.

The lack of comprehensive model clarifications of the physical procedures handling the climate system and climatic alternatives results in a high level of uncertainty in most

GCMs [27,30–32]. Utilizing as many climate models as possible is usually a good idea to account for many potential future changes [30–32]. CMIP models have been extensively refined over the years to address these uncertainties, from CMIP1 to the most recent version, CMIP6. CMIP5 was superior to CMIP3 [33–36]. The experiments and GCMs included in CMIP5 are more comprehensive and complex, and they cover a wider range of scientific problems using several representative concentration pathways (RCPs). Instead of starting with 2005 for future scenarios in CMIP5, the recently released CMIP6 has a different start year (2015), with updated emission, concentration, and land-use scenarios, known as shared socioeconomic pathways (SSPs) [33]. The CMIP6 models also more accurately represent Earth's physics [37]. The new scenarios of CMIP6 allow better impacts of climate change policy to be assessed [38]. The goal of CMIP6 is to get a deeper understanding of climate variability by conducting a series of well-coordinated experiments. CMIP6 models have greater resolution and enhanced dynamical processes [39]. Therefore, they are more accurate than previous versions [4,40]. Some studies have reported the robustness of the new CMIP6 over the CMIP5 in America [41], Asia [27,42,43], Africa [31,44], Canada [45], China [1,46], and Korea [40].

The Middle East and North African (MENA) countries have several common characteristics, including mountainous topography, distinct orography, water scarcity, and long, hot summers. Climate fluctuation is widespread in the MENA area. Dominated by an arid climate, it is the world's driest area [47]. Despite having just 6% of the world's population, the MENA region contributes 8.7% of the world's greenhouse gas emissions [48]. Most of these emissions come from the energy sector, an essential part of many economies because of their enormous oil and gas reserves. The IPCC predicts that MENA's climate difficulties will increase during the next century [49]. National and international agencies have proposed various mitigation measures. Parts of the MENA region may be uninhabitable by 2100 if the present trends continue [50]. National and international agencies have proposed various measures to mitigate the impacts. Most of them are based on RCPs using CMIP5 models. Understanding the variations in projections between CMIP5 and CMIP6 scenarios is vital to restructure the mitigation measures based on new scenarios.

This study assessed the ability of 11 CMIP5 and CMIP6 GCMs to replicate the observed climatologies over the MENA region to quantify their differences. The study assessed the difference in CMIP's projections of several climatic variables, including rainfall, wind speed, surface pressure, and temperature.

## 2. Study Region

There are 20 nations in MENA, located between 17° W–60° E and 9° N–38° N, as shown in Figure 1. It covers 13.3 million km$^2$ and is the home of nearly 500 million people. Its coastal and marine environment consists of five oceanic realms, four coastline areas, and five provinces of fauna. The mean temperature of MENA ranges from −5 to 47 °C, and the annual rainfall is between 0 and 1000 mm. Two-thirds of the region suffers from water deficiency and desertification due to high aridity. MENA countries have similar features, such as steep terrain, noticeable orography, lack of water, and scorching summers.

MENA experiences widespread climate variability. The region has six climatic zones according to the Köppen classification: tropical Savannah climate (Aw), dry arid steppe (BS), dry arid desert (BW), mild temperate fully humid (Cf), mild temperature dry summer (Cs), and mild temperature dry winter (Cw) [51]. As a result of their combined modest area coverage, all mild temperate zones are integrated to zone C (Figure 1). The BW climatic zone covers 86% of the region's total land area and has annual rainfall ranging from 0 to 100 mm. The temperature decreases to −6 °C in the winter, especially in zone C. Temperatures in the BS and BW zones frequently exceed 40 °C during the summer months. The Aw zone has monthly rainfall ranging from 0 to 200 mm, whereas the BS and BW zones have monthly rainfall peaks of 50 mm.

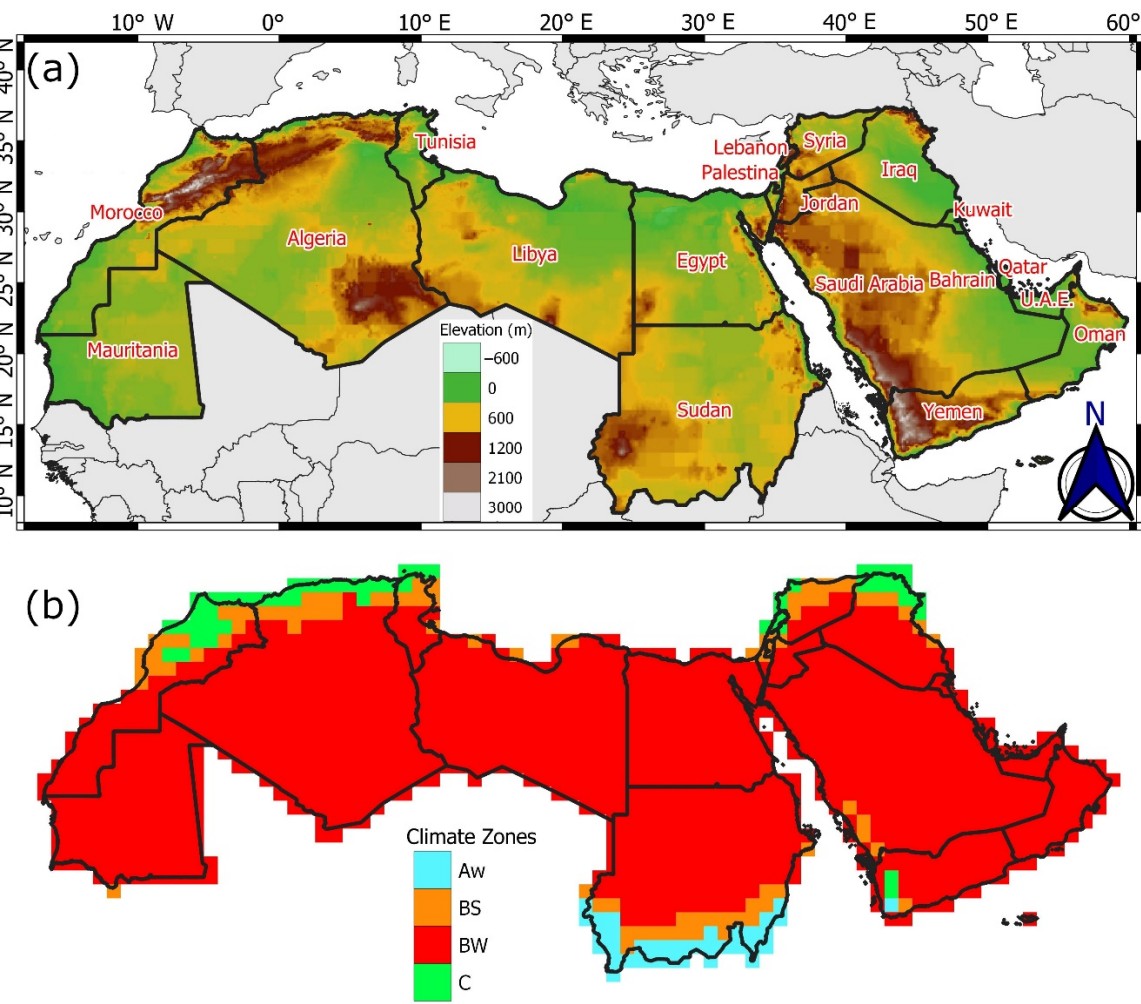

**Figure 1.** MENA (**a**) region topography and (**b**) climate zones.

### 3. Data

#### 3.1. Gridded Dataset

The ERA5 (worldwide high-resolution reanalysis dataset) was used to evaluate the GCM's ability to replicate annual rainfall, mean maximum ($T_{max}$) and minimum ($T_{min}$) temperatures, mean eastward (uas) and northward (vas) wind speed, and mean surface pressure. As part of the Copernicus Climate Change Service, the European Centre for Medium-Range Weather Forecasts (ECMWF) released the fifth edition (ERA5) of its atmospheric, oceanic, and land surface reanalysis product [52]. A total of 240 atmospheric variables are available for different pressure levels. Global observation data and an upgraded Integrated Forecasting System (IFS) cycle 41r2 were combined to produce ERA5. This research used the hourly ERA5 dataset of five meteorological variables (e.g., rainfall, surface pressure, eastward and northward wind speed, and near-surface temperature) with a 0.25-degree spatial resolution from January 1965 to December 2005. The hourly surface pressure and wind speed were used to obtain the mean monthly value, whereas the highest and lowest daily temperatures were used to extract the mean $T_{max}$ and $T_{min}$. On the other hand, the hourly rainfall data were used to obtain the total monthly rainfall.

Collecting vast amounts of observation data for a large-scale study area is difficult due to the unavailability of data or bureaucracies in the developing country. MENA is deemed a data-scarce area due to the scarcity of high-quality long-term observation data [53–56]. Evenly spaced gridded datasets are commonly utilized in data-scarce locations for model validation. Many researchers used ERA5 as a reference gridded dataset when studying the MENA region [57,58].

The geographical distribution of average $T_{max}$, $T_{min}$, surface pressure, uas, vas, and annual rainfall over the MENA region are shown in Figure 2. Most of MENA received an annual rainfall of only 0 to 100 mm, whereas the highest annual rainfall (>500 mm) occurred in the north of Iraq, Algeria, Tunisia, and Morocco (>35° N) and south of Sudan and Yemen (<15° N). The surface pressure ranged from 80 to 105 kPa, with the lowest in the north of Morocco, southwest of Saudi Arabia, and the west region of Yemen, and the highest in Iraq, Libya, Egypt, Tunisia, and Mauritania. The MENA region's $T_{max}$ uniformly varied from 25 to 35 °C. Except for the east of Sudan, Saudi Arabia and the south of Mauritania, $T_{max}$ exceeded 35 °C. The lowest $T_{max}$ (~18 °C) happened in the north of Morocco. $T_{min}$ ranged from 5 to 25 °C; however, in the north of Iraq, it may have been as low as 0 °C. Generally, Oman, the southeast of Sudan, the south of Saudi Arabia, and Mauritania experienced the highest $T_{max}$ and $T_{min}$, and the northern coastal region experienced the lowest temperatures. The wind speed ranged from −6.5 to 4.5 m/s, where most areas faced positive uas and negative vas. The highest uas was in the northeast region, whereas the highest vas was in Oman. The lowest vas was in Egypt, Sudan, and the south of Morocco.

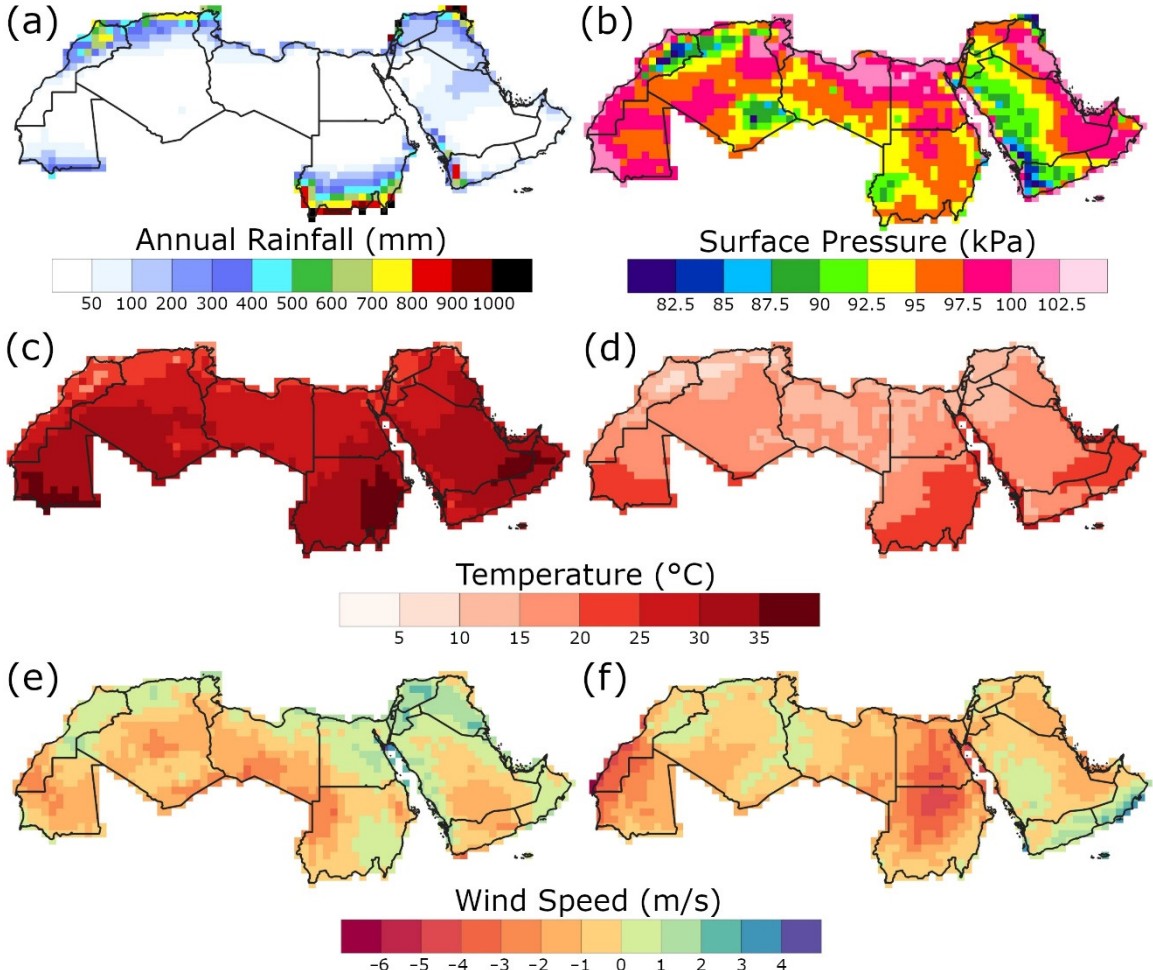

**Figure 2.** Spatial variability of (**a**) annual rainfall (mm), (**b**) annual mean surface pressure (kpa), (**c**) $T_{max}$ (°C), (**d**) $T_{min}$ (°C), (**e**) uas (m/s), and (**f**) vas (m/s) over MENA during 1965–2005, estimated via ERA5.

### 3.2. CMIP5 and CMIP6 GCMs

Climate models are most widely used for projecting the future of the Earth's climate. This study examined the historical and future projection of both CMIP5 and CMIP6 over 1219 grid points. Monthly historical (1965–2005) and future projection (2020–2099) rainfall,

$T_{max}$, $T_{min}$, surface pressure, uas, and vas were used for the assessment [29,33]. Eleven GCMs for both CMIPs in the same institutions were selected based on the availability of the mentioned variables (Table 1). The simulations can be obtained from https://esgf-node.llnl.gov/projects/esgf-llnl/ accessed on 1 July 2022. The initial variant label, r1i1p1 for CMIP5 and r1i1p1f1 for CMIP6, was selected to ease the evaluation procedure. Medium radiation scenarios (RCP4.5 and SSP2-4.5), an intermediate stabilization scenario, presuppose the implementation of emissions mitigation initiatives. On the other hand, high radiation scenarios (RCP8.5 and SSP5-8.5) represent a return to business as usual in the face of continued reliance on fossil fuels [41]. CMIP5 RCP4.5 and 8.5 scenarios are identical to CMIP6 SSP2-4.5 and SSP5-8.5 scenarios regarding radiative forcing.

**Table 1.** List of GCMs compared in the study.

| CMIP5 | | CMIP6 | | Institution | Country |
|---|---|---|---|---|---|
| Model | Resolution | Model | Resolution | | |
| ACCESS1-3 | 1.90 × 1.20° | ACCESS-CM2 | 1.87 × 1.25° | Australian Research Council Centre of Excellence for Climate System Science | Australia |
| BCC-CSM1-1-m | 2.80 × 2.80° | BCC-CSM2-MR | 1.12 × 1.12° | Beijing Climate Center | China |
| CanESM2 | 2.80 × 2.80° | CanESM5 | 2.79 × 2.81° | Canadian Centre for Climate Modelling and Analysis | Canada |
| CMCC-CM | 0.70 × 0.70° | CMCC-ESM2 | 0.94 × 1.25° | Euro-Mediterranean Centre on Climate Change coupled climate model | Italy |
| GFDL-ESM2G | 2.50 × 2.00° | GFDL-ESM4 | 1.00 × 1.25° | Geophysical Fluid Dynamics Laboratory | USA |
| INMCM4.0 | 2.00 × 1.50° | INM-CM5-0 | 2.00 × 1.50° | Institute for Numerical Mathematics | Russia |
| IPSL-CM5A-LR | 3.70 × 1.90° | IPSL-CM6A-LR | 2.50 × 1.27° | Institute Pierre Simon Laplace (IPSL) | France |
| MIROC5 | 1.40 × 1.40° | MIROC6 | 1.40 × 1.40° | Japan Agency for Marine-Earth Science and Technology (JAMSTEC) | Japan |
| MPI-ESM-LR | 1.90 × 1.90° | MPI-ESM1-2-LR | 1.87 × 1.86° | Max Planck Institute for Meteorology (MPI-M) | Germany |
| MPI-ESM-MR | 1.90 × 1.90° | MPI-ESM1-2-HR | 0.94 × 0.94° | | |
| MRI-CGCM3 | 1.10 × 1.10° | MRI-ESM2-0 | 1.12 × 1.12° | Meteorological Research Institute | Japan |

## 4. Methodology

Over the MENA region, both CMIP5 and CMIP6 historical data (1965–2005) were compared using ERA5 (0.25° × 0.25°) as a historical reference dataset. For historical evaluation, statistical and graphical measures were used. Bilinear interpolation was employed to re-grid GCMs into a common grid resolution of 1.00° × 1.00°. The bilinear interpolation provided smooth interpolated data by utilizing four points surrounding the target point [31]. Previous studies also re-gridded CMIP5 and CMIP6 GCMs to the resolution of 1.00° × 1.00° [27,31,43,59,60]. Accordingly, the ERA5 dataset was aggregated to comply with the interpolated GCM's resolution. Each GCM was evaluated independently according to its capacity to mimic MENA's observed climate. To reduce uncertainty in future climate-change simulations and better portray climate change, a mean and median multimodel ensemble (MME) was used. The most accurate MME for replicating the historical ERA5 was used to calculate the biases in historical simulation and the projections of future climates.

### 4.1. Historical Evaluation

Statistical and graphical methods were used to evaluate CMIP5 and CMIP6 GCM MME's performance for the period 1965–2005. As the statistical metric, the Kling-Gupta efficiency (KGE) was used [61,62]. KGE evaluates Pearson's correlation (r), the spatial variability ratio, and the normalized variance as a single measure, as shown in Equation (1). The triangulation of these three parameters yields crucial diagnostic data on the model's performance. When describing and quantifying the overall fitness of GCMs, KGE is better since it is less sensitive to extremes and more capable [63]. The KGE value ranges from

1 to -∞, where 1 signifies a perfect match. Based on six climatic variables, the KGE was computed for each GCM and compared to a historical dataset (1965–2005).

$$KGE = 1 - \sqrt{(r-1)^2 + \left(\frac{\mu_{GCM}}{\mu_{ref}} - 1\right)^2 + \left(\frac{\sigma_{GCM}/\mu_{GCM}}{\sigma_{ref}/\mu_{ref}} - 1\right)^2} \tag{1}$$

where $\mu_{GCM}$ and $\mu_{ref}$ are the mean, and $\sigma_{GCM}$ and $\sigma_{ref}$ are the standard deviation for GCM and ERA5 data, respectively.

The Taylor diagram was used to assess the performance of the GCMs and the mean and median MME for both CMIPs [64]. The results are presented in the Supplementary Materials. Using three statistical measures, including the centered root-mean-square error (CRMSE), degree of correlation (R), and the ratio of spatial standard deviation (SD), the figure is a robust visual representation. The two CMIPs were compared to ERA5 data in CRMSE to find inconsistencies. As you move away from the center of this figure, the value of CRMSD increases along the blue line.

### 4.2. Future Projections

Climate change in the MENA region was projected using GCMs from both CMIPs and compared to the historical era (1965–2005) for annual rainfall, $T_{max}$, $T_{min}$, surface pressure, uas, and vas. Medium- (RCP4.5 and SSP2-4.5) and high-impact (RCP8.5 and SSP5-8.5) scenarios were investigated in this study. Near (2020–2059) and far (2060–2099) futures were used to compare spatial distributions extensively. For each scenario, the projection interval's median and 95% confidence band were taken into account to calculate the corresponding uncertainty of the various CMIP models. The absolute change for each variable is presented as a map for each CMIP model.

## 5. Results

### 5.1. Historical Evaluation Skills of GCMs

Figure 3 depicts the performances of both CMIP5 and CMIP6 in replicating ERA5 annual rainfall, surface pressure, $T_{max}$, $T_{min}$, uas, and vas in terms of KGE. KGE values of both CMIP models are presented in the radar chart, where light green is CMIP5 and light red is CMIP6, along with their mean and median MME. Most CMIP6 models had greater KGEs values for all six variables, suggesting superior performance to their previous CMIP5 equivalents. CMIP6 models reproduced $T_{max}$ and surface pressure more precisely than other variables. Except for MIROC, the newer model's simulations of historical $T_{max}$ were superior to those of their older counterparts. However, CanESM, MIROC, and MPI.ESM.HR in CMIP5 were better for $T_{min}$ and BCC, CMCC, and MIROC were better for surface pressure. Besides, ACCESS, CanESM, MPI.ESM.LR, and MPI.ESM.HR in CMIP5 were better for rainfall. For wind speed components, MIROC and MPI.ESM.LR were better for uas and CMCC, IPSL.CM.LR, MIROC, and MPI.ESM.LR were better for vas. KGE for uas was less than zero for BCC, CanESM, and MRI CMIP6, whereas 5 GCMs of CMIP5 were less than zero. CMIP5 uas mean MME showed a negative KGE value (−1.00). For individual comparison, GFDL-ESM CMIP6 performed well in almost all variables. The KGE values for median MME for both CMIPs were better than mean MME in all variables, except for rainfall. At the same time, both mean and median MME CMIP6 were superior to CMIP5 in all variables. According to individual GCMs in both CMIPs, the performance of the median MME rainfall was lower than the individual GCMs.

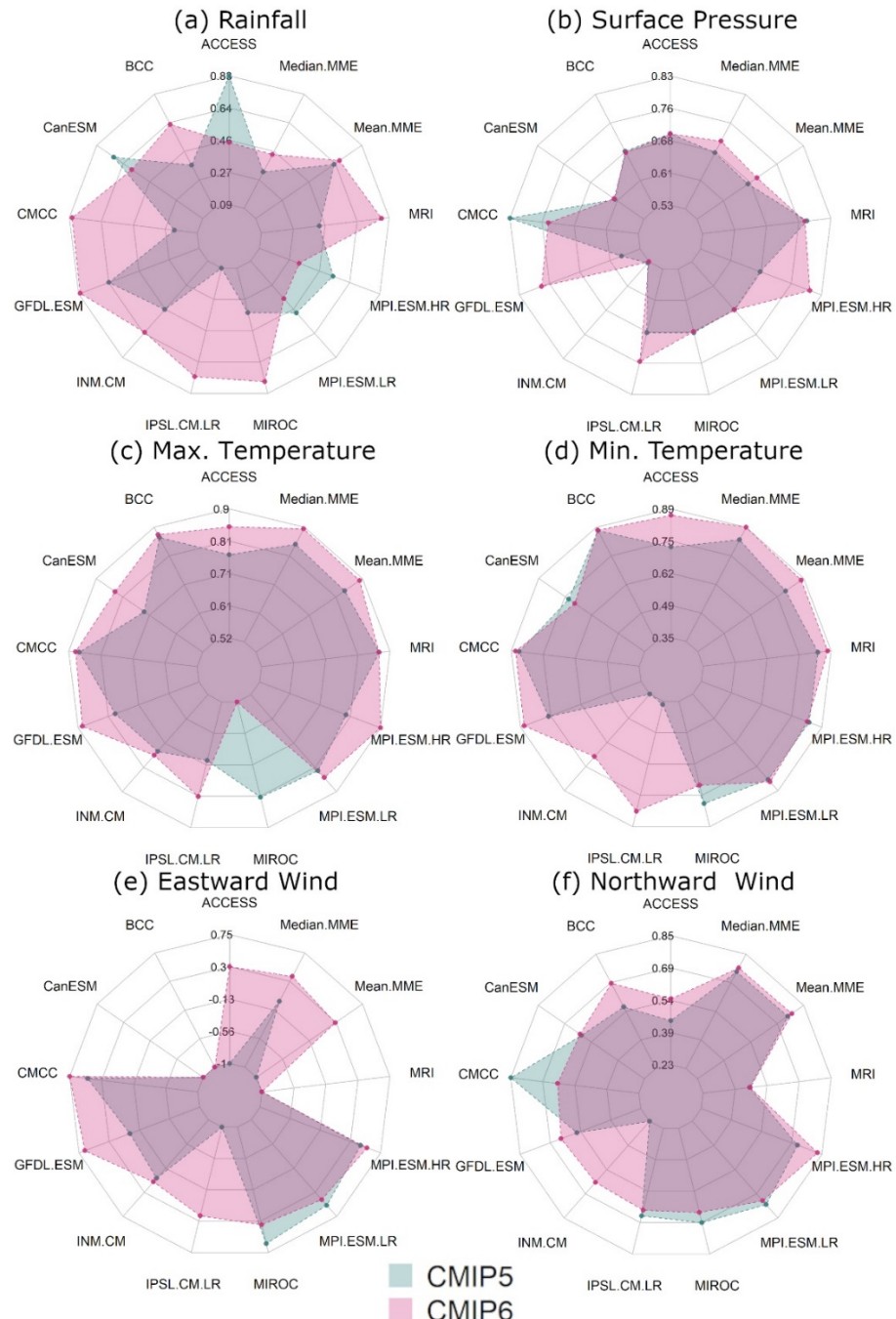

**Figure 3.** CMIP5/CMIP6 KGE statistical performance in reproducing historical (1965–2005) mean (**a**) annual rainfall, (**b**) surface pressure, (**c**) $T_{max}$, (**d**) $T_{min}$, (**e**) uas, and (**f**) vas.

For historical assessment, Taylor diagrams were used in Figure S1 to estimate the ability of both CMIP models to replicate the abovementioned variables along with their mean and median MME. ERA5 is presented as a hollow circle on the x-axis, with the GCM symbol (colored circles for CMIP5 and triangles for CMIP6) closest to the hollow circle, which signifies the highest-performing model. The CMIP6 models were closer to ERA5 than CMIP5 models. In addition, both mean and median CMIP6 MME were closer to ERA5. The median MME was chosen from radar charts and Taylor diagrams to be compared spatially with ERA5 and project the climate variables.

## 5.2. Median MME Bias Spatial Distribution

The 50th percentile of all individual GCM was used to produce the median historical (1965–2005) MMEs of CMIP5 and CMIP6. The geographical distribution of bias in GCM simulations was evaluated by comparing the MMEs to ERA5. The absolute biases between median MME and ERA5 for mean annual rainfall, surface pressure, $T_{max}$, $T_{min}$, uas, and vas for both CMIPs are presented in Figure 4. The findings indicate an improvement of CMIP6 in mimicking the geographical distribution for all variables. The overestimation of $T_{max}$ in the west was higher in CMIP6. CMIP6 overestimated $T_{min}$ in Egypt, Libya, and Sudan, whereas underestimation was less in Saudi Arabia and Algeria. $T_{min}$ bias was less in CMIP6 ($-0.11$ °C) than in CMIP5 ($-0.87$ °C). For annual rainfall and uas, CMIP6 overestimated more than CMIP5 in different locations. However, in surface pressure and vas, CMIP6 underestimated in different locations than CMIP5.

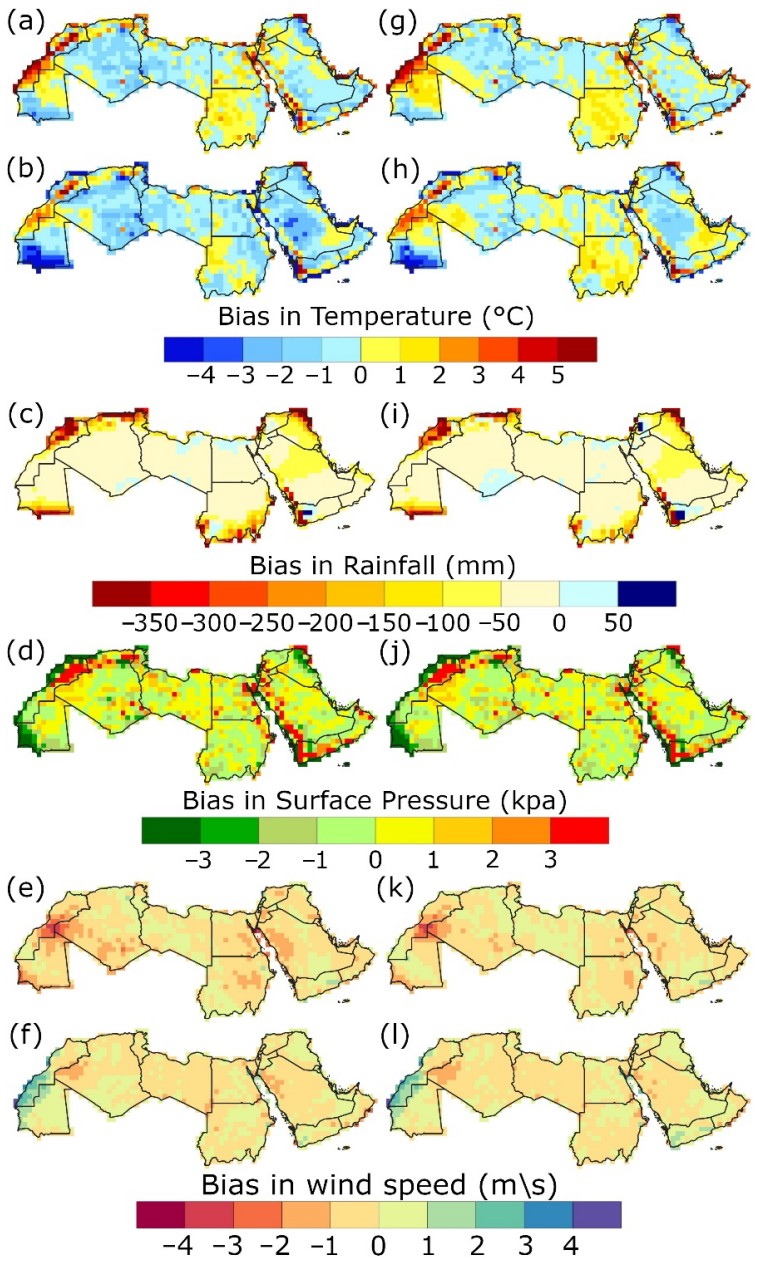

**Figure 4.** Bias between median MME of CMIP5 (left column) and CMIP6 (right column) for $T_{max}$ (**a**,**g**), $T_{min}$ (**b**,**h**), annual rainfall (**c**,**i**), surface pressure (**d**,**j**), uas (**e**,**k**), vas (**f**,**l**), and ERA5.

### 5.3. Seasonal Variability

The seasonal variability of surface pressure, temperature, rainfall, and wind speed in each climatic zone was compared to ERA5 using the median MME of the 11 available GCMs of each CMIP for the historical period. Median $T_{max}$ (left panel) and $T_{min}$ (right panel) month-to-month variability of CMIP5 and CMIP6 estimations are shown in Figure 5 as dashed red and dashed blue lines, respectively, compared to the solid black line of ERA5. It also shows the 95% confidence interval (CI) monthly estimates to provide the degree of uncertainty and model variability. The temperature pattern was different in the Aw zone compared to other climatic zones. Overall, the $T_{max}$ CI band of CMIP6 was wider than that of CMIP5, whereas in $T_{min}$ the CMIP6 band was much thinner than that of CMIP5. Both CMIP models overestimated $T_{max}$ in all zones, especially in summer. The highest overestimation was for zone C. CMIP6 simulations showed a substantially smaller uncertainty in $T_{min}$ than in CMIP5. $T_{min}$ was overestimated in Aw and C climatic zones by GCMs of both CMIPs for most months. The $T_{min}$ medians of both MMEs are almost equal to ERA5 $T_{min}$ for the climatic zones BS and BW.

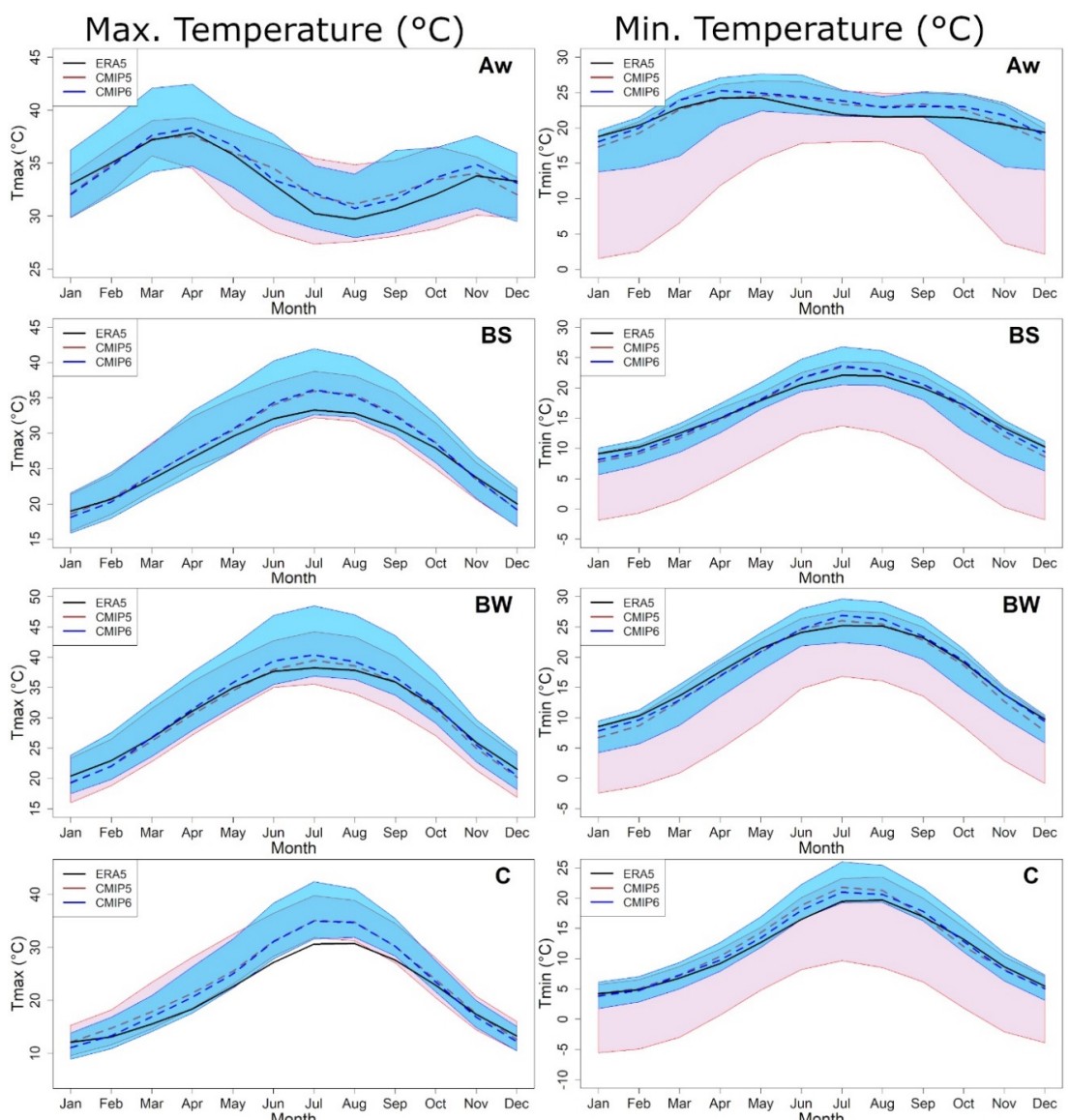

**Figure 5.** Seasonal variability in mean $T_{max}$ and $T_{min}$ of CMIP5 and CMIP6 GCMs compared to ERA5 in different climate zones (Aw, BS, BW, and C) of MENA.

The CMIP6 rainfall uncertainty band was narrower for all climatic zones than CMIP5, as shown in Figure 6. Both CMIPs underestimated month-to-month rainfall in most months, except in the BW zone. Rainfall in zone C was the most underestimated by both CMIPs. The CMIPs' median MME and CI bands were below ERA5 for most of the year. This suggests that all GCMs in this zone had underestimated rainfall. Both bands were similar in the Aw zone for surface pressure, whereas CMIP6 was wider for the BS and BW zones and lesser for zone C. Overall, the CMIP6 median was closer to ERA5 in all zones, indicating the better performance of CMIP6. Both CMIPs overestimated month-to-month surface pressure during most months, except in zone BW.

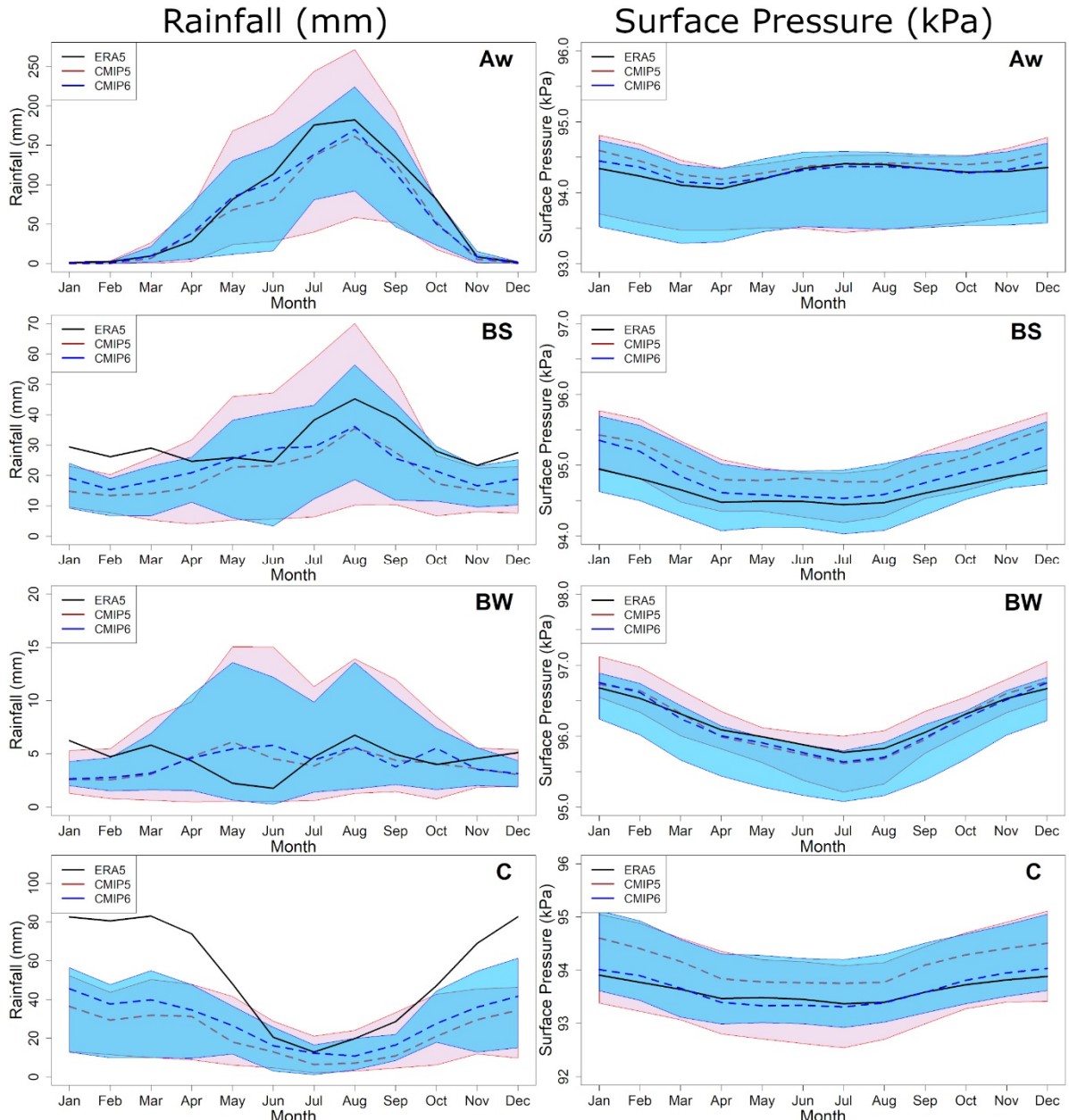

**Figure 6.** Same as Figure 5, but for rainfall (left column) and surface pressure (right column).

Figure 7 presents the month-to-month variability of both CMIPs for uas (left panel) and vas (right panel). Overall, the CMIP6 CI band was thinner than that of CMIP5 and the CMIP6 median was closer to ERA5 in all zones for both variables. For the Aw zone, both models overestimated ERA5 uas and vas in summer and underestimated them in winter.

For zones BS and BW, both CMIPs underestimated those variables, except in summer for vas. Summer experienced an underestimation in uas in zone C, whereas both CMIPs were closer to ERA5 in vas.

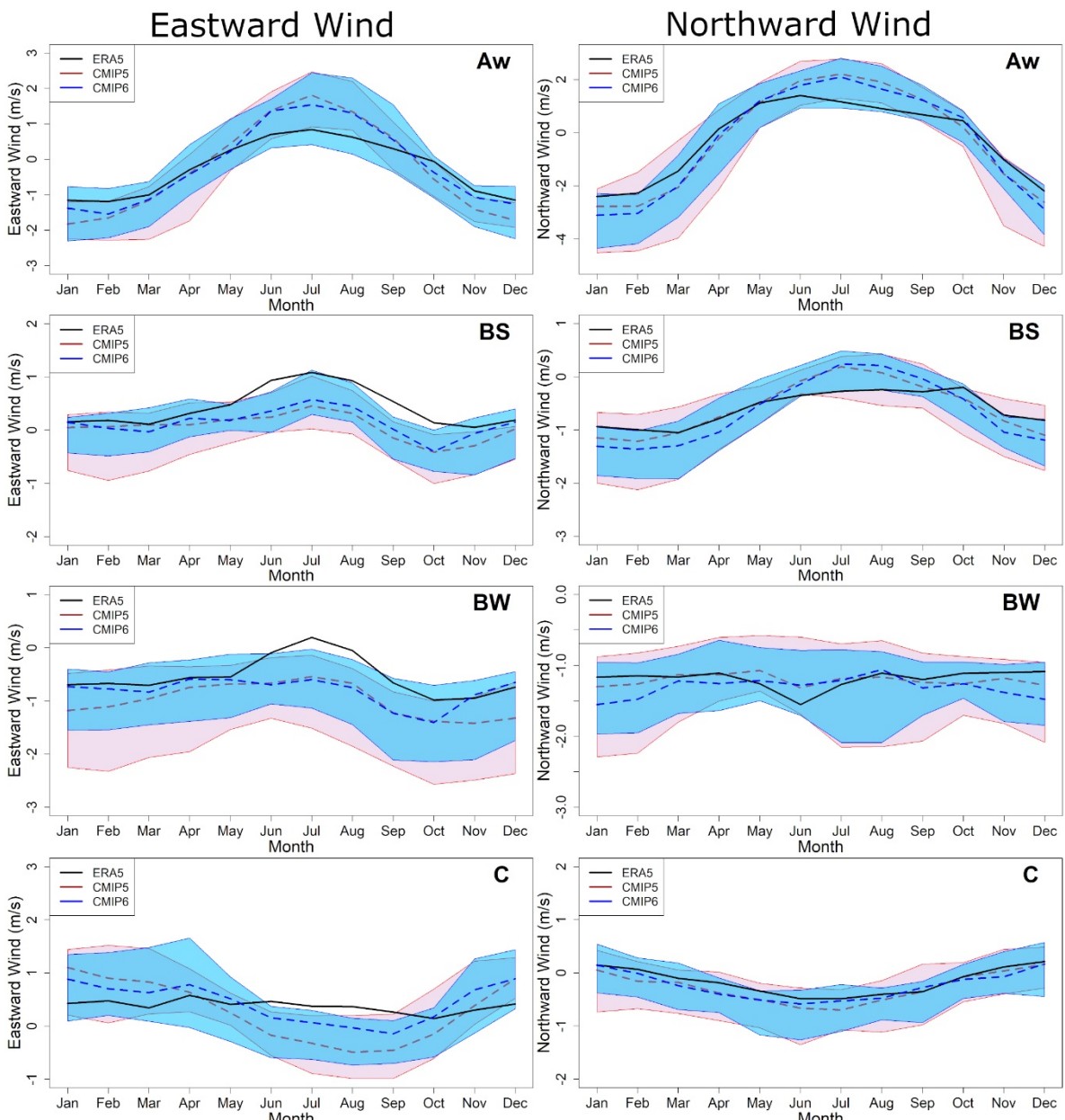

**Figure 7.** Same as Figure 5, but for eastward wind (left column) and northward wind (right column).

### 5.4. Annual Variable Projection

Temporal analysis of GCM $T_{max}$ (left panel) and $T_{min}$ (right panel) from 1965 to 2100 over the MENA are presented in Figure 8. The top plots (a and c) show the projections for medium (RCP4.5 for CMIP5 and SSP2-4.5 for CMIP6) emission scenarios, whereas the bottom plots (b and d) show the projections for high (RCP8.5 for CMIP5 and SSP5–8.5 for CMIP6) emission scenarios. Both CMIP models are presented using a solid line for the historical and a dashed line for the future to represent the median MME along with their 95% CI band. CMIP6 is represented by the blue line, whereas CMIP5 is represented by the brown line, estimated using moving averages of 30 years.

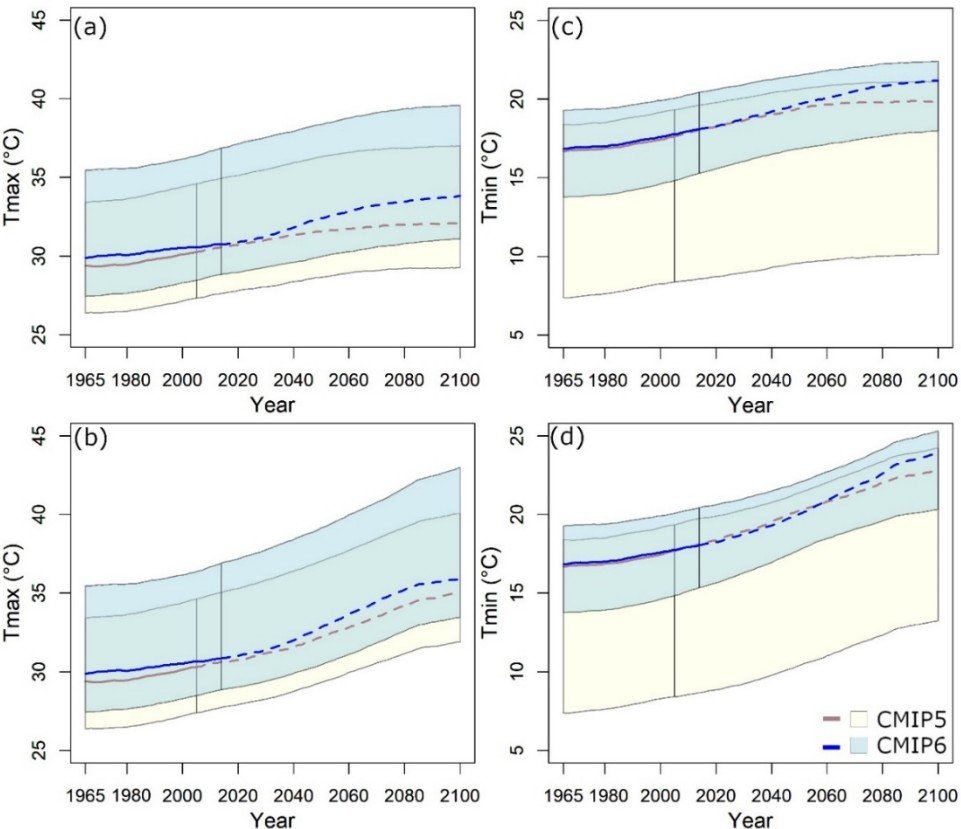

**Figure 8.** Temporal evolution of annual mean $T_{max}$ (°C) (**a**,**b**) and $T_{min}$ (°C) (**c**,**d**) for CMIP5 (yellow) and CMIP6 (blue) for different scenarios: (upper row) RCP4.5 and SSP2-4.5 and (lower row) RCP8.5 and SSP5-8.5. Shadings signify 95% projection confidence intervals. The vertical line indicates the end of the historical estimations.

The CI band was similar for both CMIPs for $T_{max}$ with an upward shift of CMIP6, whereas the CMIP6 band was much thinner for $T_{min}$, indicating less uncertainty. The CMIP6 median line for $T_{max}$ was higher than CMIP5 for both scenarios. The median value of $T_{max}$ would rise to 33.83 °C and 35.90 °C for SSP2–4.5 and 5–8.5 and 32.09 °C and 35.13 °C with RCP4.5 and 8.5, respectively, by 2100. CMIP6 band showed a sharper rate of increase, especially for SSP5-8.5. For $T_{min}$, both CMIP medians were identical from 1965 to 2015 for both scenarios. RCP4.5 for $T_{max}$ and $T_{min}$ showed a constant median value from 2050 to 2100. SSP5-8.5 showed a minor increase from 2015 to 2060 compared to RCP8.5, whereas SSP5-8.5 showed a sharp increase. Median $T_{min}$ would rise to 21.17 °C and 23.92 °C for SSP2–4.5 and 5–8.5, and 19.81 °C and 22.80 °C for RCP4.5 and 8.5, respectively. The temporal evolution of mean summer and winter $T_{max}$ and $T_{min}$ over the MENA region from 1965 to 2100 are presented in Figure S2 in the Supplementary Files. It provided similar results for the annual timescale.

Figure 9 shows the historical and future evaluation of uas (left panel) and vas (right panel) for the medium and high scenarios for the two CMIPs. Overall, the CMIP6 CI band was thinner than CMIP5, indicating less uncertainty in CMIP6. The CI band of both scenarios in CMIP6 showed an upward shift in uas. The median line was higher for CMIP6 uas and lower for vas for both scenarios. All median lines in Figure 9 showed no change in the future.

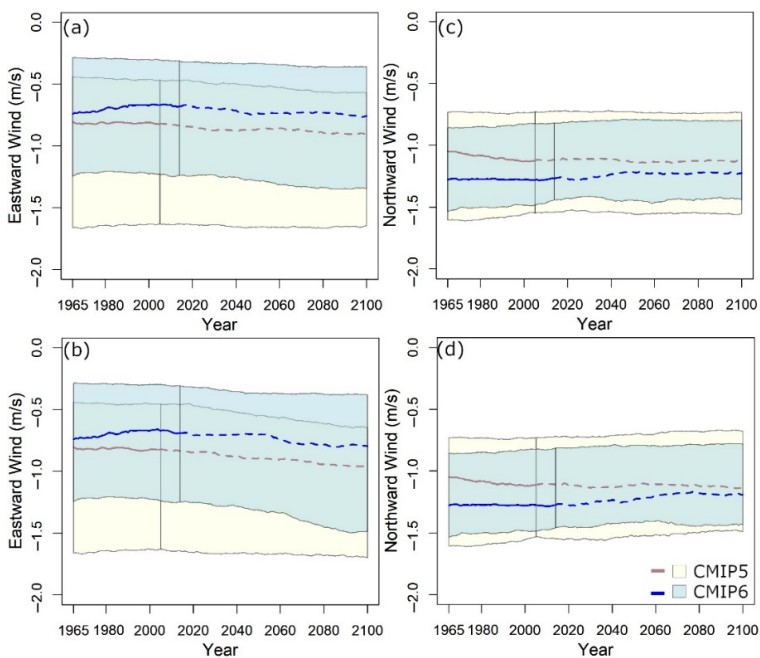

**Figure 9.** Same as Figure 8, but for uas (m/s) (**a**,**b**) and vas (m/s) (**c**,**d**).

Rainfall amounts projected by both CMIP5 and CMIP6 MME models are shown in Figure 10. It appears that the median line of CMIP6 implied a larger likelihood of more rainfall in the future than CMIP5. CMIP6 projections showed a substantially narrower CI (less uncertainty) than CMIP5. Results indicated a more significant increase in rainfall for higher scenarios than lower scenarios. CMIP5 showed a decrease in annual rainfall by 1.50 mm for RCP4.5 and an increase by 5.00 mm for RCP8.5. In contrast, it showed an increase in rainfall by 14.50 mm and 28.00 mm for SSP2-4.5 and SSP5-8.5. The right panel of Figure 10 shows the projected surface pressure in kPa for median and high scenarios. In the historical period, the CMIP6 median line was lower than CMIP5 for both scenarios, whereas in the future, both median lines were constant with a 96 kPa for both scenarios.

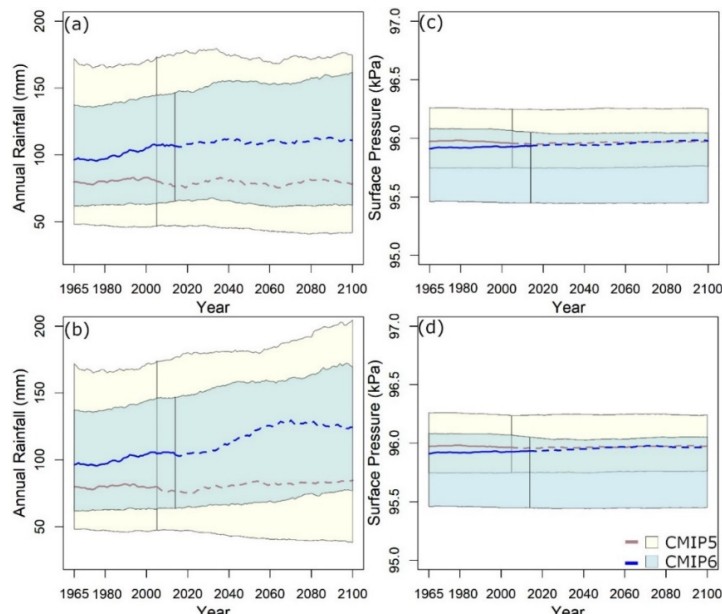

**Figure 10.** Same as Figure 8, but for rainfall (mm) (**a**,**b**) and surface pressure (kPa) (**c**,**d**).

### 5.5. Spatial Changes of Both CMIP5 and CMIP6

The spatial distribution of climate variable changes for both the near (2020–2059) and far (2060–2099) futures compared to the reference period (1965–2005) are presented in Figures 11–13. The changes were calculated using the median MME of each CMIP for both medium (RCP4.5 and SSP2-4.5) and high (RCP8.5 and SSP5-8.5) scenarios.

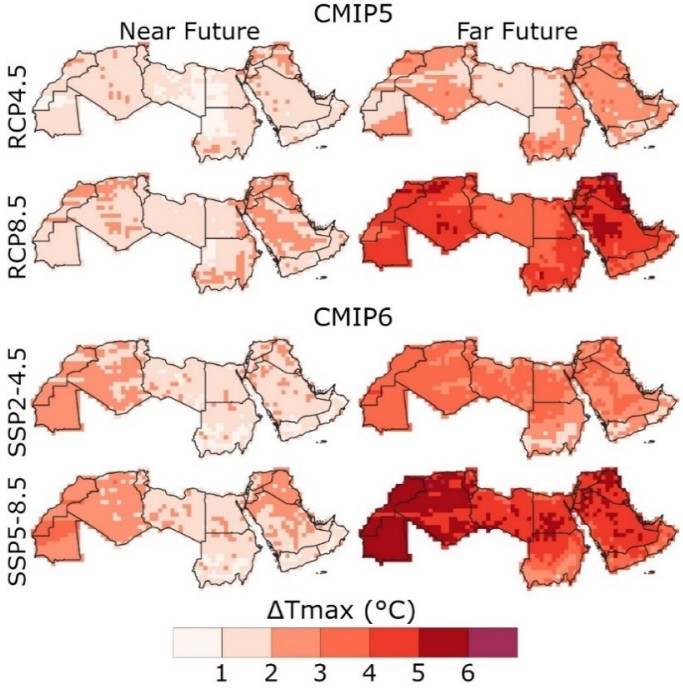

**Figure 11.** Geographical variability of the change in $T_{max}$ (°C) over MENA based on CMIP5 and CMIP6 MME in two future periods—near future (2020–2059) and far future (2060–2099)—for the medium and high projection scenarios.

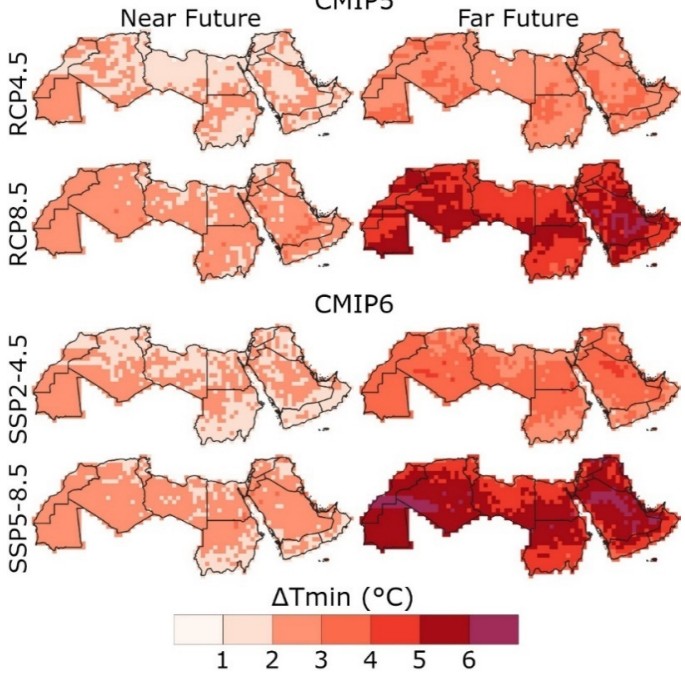

**Figure 12.** Same as Figure 11, but for $T_{min}$ (°C).

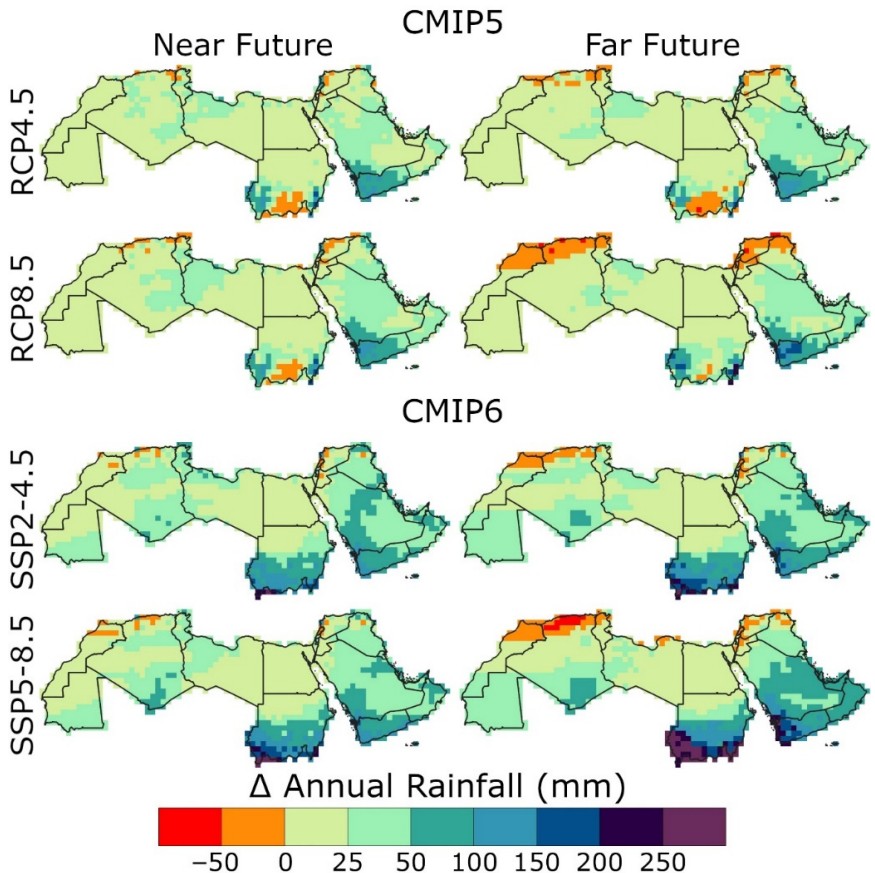

**Figure 13.** Same as Figure 11, but for annual rainfall in mm.

Figure 11 shows the projected change in $T_{max}$ (°C), where the absolute change ranged from 0 to 6 °C. Median MME for CMIP6 projected a higher temperature rise than median MME for CMIP5 did. The most vulnerable regions for $T_{max}$ increase were the eastern (Iraq and Saudi Arabia) and western (Algeria, Morocco, and Mauritania) regions. The minimum increase was observed in Yemen, Sudan, Libya, and Oman. Egypt faced a higher rate of temperature increase in CMIP6 compared to CMIP5. The projected temperature difference between CMIP6 and CMIP5 was 0.36 °C and 0.84 °C for the medium scenario, and 0.25 °C and 0.39 °C for the high scenario in the near and far future, respectively.

$T_{min}$ projected a greater increase for both CMIPs MME in the two scenarios and future periods, as shown in Figure 12. Both the northern coastal and southern regions (south of Sudan, Yemen, and Oman) faced a lower increase in $T_{min}$ than the central region of the study area. The projected increase in $T_{min}$ was higher than that of $T_{max}$ by 0.63 °C and 0.64 °C for RCP8.5, whereas the increase was 0.29 °C and 0.57 °C for SSP5-8.5 in the near and far futures, respectively. $T_{min}$ reached the highest increase of 6.5 °C in Saudi Arabia and Algeria in the SSP5-8.5 far future.

Figure 13 describes the annual rainfall spatial changes over the MENA region in mm for median MME of CMIP5 and CMIP6. Overall, MENA annual rainfall was projected to increase up to 270 mm in the south of Sudan for CMIP6, and decrease ($<-50$ mm) in the northern coastal region and south of Sudan for CMIP5. The decrease was projected in the north of Syria, Iraq, Algeria, and Morocco, with a higher rate in the north of Algeria. Egypt showed a lower increase ($<25$ mm), except in the northern coastal region.

The spatial changes for surface pressure, uas and vas, are shown in Figures S4–S6. The surface pressure change ranged from $-150$ to 150 Pa over the MENA region. The negative change was concentrated in Saudi Arabia, Egypt, Iraq, and central Morocco, and the positive change was concentrated in Yemen, Algeria, north of Morocco, and south of Sudan. The change in the far future was more intense than in the near future.

The projected change for the eastward wind speed ranged from −0.8 to 0.75 m/s, with the highest increase in Sudan and Mauritania and the highest decrease in Algeria, Morocco, and western Libya. On the contrary, the greatest change in northward wind speed was in Saudi Arabia, Oman, and Yemen, whereas Egypt and eastern Libya showed the lowest change.

## 6. Discussion

CMIP5 and CMIP6 GCMs have been the subject of research to estimate their capacity to replicate historical climate in various parts of the world [27,31,42,57,65–69]. Overall, the previous studies showed that the CMIP6 models are better than their predecessors in CMIP5. The study also concluded that CMIP6 was better for simulating climate variables with lower uncertainty. For instance, Hamed et al. [31] found that CMIP6 had improvements over CMIP5 in annual rainfall and temperature simulation over Egypt. They concluded that $T_{max}$ could be replicated more accurately than $T_{min}$ using CMIP6. In addition, Zamani et al. [42] reported the better performance of CMIP6 in replicating rainfall over Iran. However, the previous studies were conducted mostly for rainfall and temperature. The relative performance of CMIP5 and CMIP6 GCMs in other simulated climate variables, like surface pressure and wind speed, is less studied. There is no such study on the MENA region.

This study assessed the performance of CMIP5 and CMIP6 GCMs' rainfall, $T_{max}$, $T_{min}$, uas, vas, and surface pressure using statistical measures and visual interpretation. Overall, CMIP6 showed improvement in mimicking all variables compared to CMIP5. The KGE of the GCMs suggests the superior performance of CMIP6 for all six variables, especially $T_{max}$ and surface pressure. CMIP6 models also showed higher accuracy in producing seasonal variability of climatic variables. The major improvement in CMIP6 models was in uncertainty in projections. The 95% CI band of the projections of all six climate variables was much narrower for CMIP6 than CMIP5. This indicates higher reliability in climate projections in MENA using CMIP6 models.

Relative assessment of GCM performance revealed higher skill of GFDL-ESM of CMIP6 in reproducing all variables over MENA. However, a single GCM is not suggested for climate change projection, as it cannot provide uncertainty in projections. Therefore, the present study suggests MME of the GCM ensemble for climate projections. The median MME of CMIP GCMs showed better performance than the mean MME of the GCMs. Therefore, the median MME of CMIP6 GCMs should be used for MENA climate projections and impact assessment.

In this research, several GCMs were used, each with a unique spatial resolution. To create an ensemble, they were all re-gridded to a common resolution of 1°. Different researchers used different resolutions for GCM comparison and climate projection [70–72]. Re-gridding of GCMs to high resolution adds uncertainty to the simulations. Therefore, most previous studies used 1° resolution, which is the mean resolution of all CMIP6 GCMs, for comparison [5,11,60,73–75]. Therefore, this analysis also adhered to the standard resolution of 1° used in most GCM performance comparison studies. CMIP5 and CMIP6 GCM projections have been shown at $1.0° × 1.0°$ in previous studies in different regions, including South America [59], Egypt [31], southeast Asia [27], East Asia [43], Iran [60], and the South Pacific Oscillation [67].

Rainfall and temperature are the most widely used climate variables for climate-change impact assessment. However, other climate variables, like wind speed and surface pressure, are required along with temperature to analyze different variables, like evapotranspiration. Changes in those variables due to climate change are little known. Therefore, assessment of the models' performance in replicating those variables is important. The present study revealed that CMIP6 GCMs' median MME can be used reliably for a broad range of impact assessments.

## 7. Conclusions

The present study assessed the ability of 11 CMIP5 and CMIP6 GCMs. The relative performance of the CMIPs was evaluated based on their ability to simulate six climate variables and uncertainty in their future projections over the MENA region. This was the first attempt to assess the relative performance of CMIP5 and CMIP6 GCMs over MENA in terms of multiple climate variables. The study revealed the significantly better performance of CMIP6 GCMs than CMIP5 GCMs in simulating climate over the MENA region. The CMIP6 GCMs' MME projections were less uncertain than their counterpart in CMIP5. The results indicate reliable climate projections using CMIP6 GCMs. The present study also suggests using the MME median rather than MME mean for climate change projections. The capability of CMIP6 GCMs in simulating six climate variables indicates the applicability of the models in a wide range of climate applications. This study used 11 GCMs of CMIP5 and CMIP6 for comparison, as projections of six climate variables were available only for those 11 GCMs. This work has several limitations, but those may be considered viable for future research. In the future, the study can be repeated with more GCMs when available. Besides, the GCMs' performance in replicating dew point and solar radiation can be evaluated in the future. Future studies can be conducted to assess uncertainty in climate simulations due to re-gridding in different resolutions.

**Supplementary Materials:** The following supporting information can be downloaded at: https://www.mdpi.com/article/10.3390/su141610375/s1, Figure S1 Taylor diagrams, showing skill of the GCMs of two CMIPs in simulating: (a) annual total rainfall; and annual mean (b) surface pressure, (c) $T_{max}$, (d) $T_{min}$, (e) uas and (f) vas, Figure S2 Temporal evolution of mean summer and winter $T_{max}$ (°C) (a and b) and $T_{min}$ (°C) (c and d) for CMIP5 (yellow and red) and CMIP6 (cyan and blue) under different scenarios (upper row) RCP4.5 and SSP2-4.5 and (lower row) RCP8.5 and SSP5-8.5. Shadings signify 95% projections confidence interval. The vertical line indicates the end of the historical estimations, Figure S3 Temporal evolution of mean seasonal summer rainfall (mm) (a and b) and winter rainfall (mm) (c and d) for CMIP5 (yellow) and CMIP6 (blue) under different scenarios (upper row) RCP4.5 and SSP2-4.5 and (lower row) RCP8.5 and SSP5-8.5. Shadings signify 95% projections confidence interval. The vertical line indicates the end of the historical estimations, Figure S4 Geographical variability of the change in surface pressure (Pa) over SEA based on MME of CMIP5 and CMIP6 for two futures in medium and high projection scenarios, Figure S5 Geographical variability of the change in eastward wind speed (m/s) over SEA based on MME of CMIP5 and CMIP6 for two futures in medium and high projection scenarios, Figure S6 Geographical variability of the change in northward wind speed (m/s) over SEA based on MME of CMIP5 and CMIP6 for two futures in medium and high projection scenarios.

**Author Contributions:** Conceptualization, M.M.H.; Data curation, M.S.N.; Formal analysis, M.S.S.; Methodology, M.M.H. and M.S.N.; Software, M.S.N. and S.S.; Supervision, S.S.; Validation, M.S.S.; Visualization, M.M.H.; Writing—original draft, M.M.H., M.S.N., M.S.S. and S.S.; Writing—review & editing, S.S. All authors have read and agreed to the published version of the manuscript.

**Funding:** This research received no external funding.

**Institutional Review Board Statement:** Not applicable.

**Informed Consent Statement:** Not applicable.

**Data Availability Statement:** Not applicable.

**Conflicts of Interest:** The authors declare no conflict of interest.

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
