# Peer review of "Comparison between CMIP5 and CMIP6 Models over MENA Region Using Historical Simulations and Future Projections"

_sustainability, doi:10.3390/su141610375_

Round 1
Reviewer 1 Report
OVERVIEW
The MS refers to the topic of climate change impacts. The authors use the CMIP5 and CMIP6 datasets to estimate the reproducibility and uncertainty of various meteorological variables. The present study assessed the ability of 11 CMIP5 and CMIP6 GCMs. The relative performance of the CMIPs was evaluable based on their ability to simulate six climate variables and uncertainty in their future projections over the MENA region. The results indicate reliable climate projections using CMIP6 GCMs. The present study also suggests using the MME median rather than MME mean for climate change projections.
The MS is of interest to the Sustainability readership and is written in an acceptable manner. I have some general and specific comments, and the answers will help to better understand the details of this study.
GENERAL COMMENTS
[1] What is the physical meaning of the reduced resolution of ERA5? GCMs data are simply interpolated on a 1°x1° grid, and this approach knowingly reduces the accuracy of the reanalysis, thereby compensating for the errors inherent in GCMs. In my opinion, it would be more promising to bring the resolution of GCMs data to ERA5 (0.25°x0.25°). The results of such studies depend very much on it. Perhaps it should be mentioned in the discussion section.
[2] A lot of work has been done to get the generally trivial and expected conclusion that CMIP6 showed improvement in mimicking all variables compared to CMIP5. What constitutes the uncertainty of the projections for the region under study? Is it more important to choose more accurate models over the historical period, or to choose a more appropriate RCP/SSP scenario? The latter can be done by evaluating the dynamics of meteorological variables over the scenario period for CMIP5 (since 2006) and for CMIP6 (since 2015) relative to ERA5 up to 2021.
[3] The RCP/SSP8.5 scenario is more applicable for estimating extreme indicators (e.g., Tmin, Tmax). At the same time, changes in seasonal and especially annual average values are very overestimated. What do you think about it? How much emphasis should be placed on this scenario?
[4] The authors' recommendations are limited to only two sentences without any references.
"In the future, the study can be repeated with more GCMs when available. Besides, the GCMs’ performance in replicating dew point and solar radiation can be evaluated in the future."
Surely there are already examples for parts of the world where more than eleven models have been evaluated to improve climate prediction performance based on CMIP6.
SPECIFIC COMMENTS
[1] Line 41 GHG – transcript required
[2] Far future period is 2020-2099 (line 193). Is that correct?
[3] Links are not displayed in several places in the text: lines 82, 93, 125, 201, 237, 252, 264, 272, 284, 287, 291, 318, 324, 338, 344, 347, 357, 363, 385, 388
[4] Line 390 repeat "estimate" - research to estimate their capacity to estimate historical climate
Reviewer 2 Report
The assessment of the improvement of the Coupled Model Intercomparison Project Phase 6 (CMIP6) over Coupled Model Intercomparison Project Phase 5 (CMIP5) for temperature, wind and precipitation simulation in different regions of the world is of great interest, from both theoretical and practical viewpoints. In its context, the peer-reviewed manuscript is relevant. The authors focused their efforts on the analysis of data of CMIP5 and CMIP6 projects in relation to the North African region and Arabian Peninsula area.
In general, this study looks quite correct, but for publication, the authors need to slightly update the paper.
My comments are as follows:
1. The table with list of general circulation models (GCMs), both CMIP5 and CMIP6 should be presented. This table should include at least the space resolution and the institution in which particular model was developed.
2. The authors must describe indicators recommended by the World Meteorological Organization to monitor changes in extremes on temperature and precipitation data. Authors should then justify their choice of indicators.
3. The analysis is not presented completely. It is required to detect and present the linear trends using for example the Sen’s slope estimator, The Mann-Kendal trend test or some others.
